# Establishment and Evaluation of Atmospheric Water Vapor Inversion Model Without Meteorological Parameters Based on Machine Learning

**DOI:** 10.3390/s25020420

**Published:** 2025-01-12

**Authors:** Ning Liu, Yu Shen, Shuangcheng Zhang, Xuejian Zhu

**Affiliations:** College of Geology Engineering and Geomatics, Chang’an University, Xi’an 710054, China; ningliu@chd.edu.cn (N.L.); shuangcheng369@chd.edu.cn (S.Z.); 2022226026@chd.edu.cn (X.Z.)

**Keywords:** ground-based GNSS, random forest algorithm, atmospheric precipitable water, accuracy assessment, spatial and temporal characteristics of water vapor

## Abstract

Precipitable water vapor (PWV) is an important indicator to characterize the spatial and temporal variability of water vapor. A high spatial and temporal resolution of atmospheric precipitable water can be obtained using ground-based GNSS, but its inversion accuracy is usually limited by the weighted mean temperature, Tm. For this reason, based on the data of 17 ground-based GNSS stations and water vapor reanalysis products over 2 years in the Hong Kong region, a new model for water vapor inversion without the Tm parameter is established by deep learning in this paper, the research results showed that, compared with the PWV information calculated by the traditional model using Tm parameter, the accuracy of the PWV retrieved by the new model proposed in this paper is higher, and its accuracy index parameters BIAS, MAE, and RMSE are improved by 38% on average. At the same time, the PWV was inverted by radiosonde data in the study area as a reference to verify the water vapor inversion results of the new model, and it was found that the BIAS of the new model is only 0.8 mm, which has high accuracy. Further, compared with the LSTM model, the new model is more universal when the accuracy is comparable. In addition, in order to evaluate the spatial and temporal variation characteristics of the atmospheric water vapor retrieved by the new model, based on the rainstorm event caused by typhoon in Hong Kong of September 2023, the ERA5 GSMaP rainfall products and inverted PWV information were comprehensively used for analysis. The results show that the PWV increased sharply with the arrival of the typhoon and the occurrence of a rainstorm event. After the rain stopped, the PWV gradually decreased and tended to be stable. The spatial and temporal variation in the PWV have a strong correlation with the occurrence of extreme rainstorm events. This shows that the PWV inverted by the new model can respond well to extreme rainstorm events, which proves the feasibility and reliability of the new model and provides a reference method for meteorological monitoring and weather forecasting.

## 1. Introduction

Precipitable water vapor is the total water vapor content of the vertical column per unit area in the atmosphere, and as a key factor in regulating the earth’s climate, it plays an important role in atmospheric radiation, water cycle, and energy balance. The spatial and temporal variation in water vapor are important driving forces of climate change, so it is of important significance to understand the spatial and temporal variation in water vapor for the prediction of weather and climate evolution [1,2,3]. The traditional methods of obtaining atmospheric water vapor mainly include radiosonde and microwave radiometers. Although the PWV obtained by radiosonde has high precision, it is difficult to meet the needs of extreme weather monitoring and forecasting at medium and small spatial and temporal scales due to its high cost and low efficiency, while a microwave radiometer is often limited to small-scale areas. Therefore, ground-based GNSS water vapor inversion technology is widely used in atmospheric water vapor monitoring due to its advantages of high spatial and temporal resolution, high precision, and low cost [4].

The inversion of the PWV by ground-based GNSS requires the acquisition of zenith tropospheric wet delay (ZWD) and conversion coefficient (Π) [5,6]. The ZWD parameter can be calculated by using the zenith tropospheric delay (ZTD) minus the zenith hydrostatic delay (ZHD), and the coefficient Π requires the important factor of weighted mean temperature (Tm). Many studies have shown that Tm is one of the largest error sources in the process of calculating the PWV, and Tm usually needs to be calculated by the measured temperature and pressure data from the station, which leads to the limitation of the timeliness of the PWV inversion [7,8,9]. However, due to the lack of real measured meteorological data, the accuracy of the PWV obtained by the existing Tm calculated based on empirical model is not high; therefore, the construction of a high-precision PWV inversion model without Tm parameters and with strong timeliness is worthy of further study. In order to achieve this goal, Li used ZTD, Ts, and Ps to establish the PWV conversion model, and analyzed the accuracy of a single factor, two factors and three factors, respectively. The results show that the accuracy of the PWV inversion increases with the increase in factors, and the accuracy of the three factors is higher than that of the traditional model [10]. Taking into account the annual and spatial changes in PWV, Jiang used ERA5 data to establish a CZP model that directly converts ZTD into PWV and verified that its accuracy is higher than that of the traditional model [11]. Most of the above studies used linear regression fitting methods to model the target parameters, without considering the nonlinear relationship between the target parameters and the impact factors. Therefore, there are problems such as insufficient exploration of the functional relationship between the target parameters and the impact factors. Aiming at the above problems, many scholars have utilized intelligent algorithms to improve the accuracy of the conversion model. Huang used a deep neural network (DNN) to establish the PWV conversion model and verified the feasibility of the model [12]. Xie used ZTD, time, and other parameters to establish the PWV conversion model based on the combined genetic algorithm, its results showed that the combined model was better than the traditional model [13]. Although the above research utilized intelligent algorithms to reduce the nonlinear error of the model, their computational processes are complicated. In summary, taking into account the ERA5 reanalysis data, ground-based GNSS data and machine learning algorithms are used to establish an atmospheric water vapor inversion model without Tm in this paper, and its structure is simple and easy to implement. The PWV calculated by new model was compared with the PWV by the traditional method, and the inversion accuracy of the new model was verified. Further, compared with long short-term memory (LSTM), the new model makes up for the lack of LSTM’s need to input recent data, and the accuracy is comparable. Finally, based on two rainfall data products, the hourly PWV results and rainfall data were used to respond to the extreme rainstorm events in Hong Kong in September 2023, and the spatial and temporal characteristics of the PWV in the rainstorm process were analyzed.

## 2. Methods and Dataset

In this section, the commonly used water vapor conversion model and the method of calculating PWV using ERA5 data are introduced in detail. At the same time, the establishment process of the new model is described in detail. Subsequently, the dataset used in this paper is described.

### 2.1. PWV Inversion-Based GNSS

When a GNSS signal passes through the neutral atmosphere, the propagation of the signal will become slower and the propagation path will be curved due to the influence of the atmospheric refraction effect. At the receiver end, the signal propagation time becomes longer. This effect is usually referred to as *ZTD*, which is mainly composed of *ZHD* and *ZWD*, and it is closely related to changes in the ground climate, atmospheric pressure, temperature, and humidity. Among them, *ZHD* is relatively stable, accounting for about 90% of *ZTD*. At present, the commonly used calculation method is the Saastamoinen model, and the calculation accuracy can reach the millimeter level [14,15]. The calculation formula is as follows:(1)$$\mathrm{ZHD}=\frac{0.00227\;\times \;P}{\left(1-0.00266\;\times \;\cos\left(2\times \varphi \right)-0.00028\times H\right)}$$
where *P* is the surface pressure, and the unit is hPa; *φ* is the latitude of the station, and the unit is rad; *H* is the geodetic height of the station, and the unit is km. *ZWD* can only obtain centimeter-level accuracy by using empirical model, so *ZTD* is usually used to subtract *ZHD*. The calculation formula is as follows:(2)ZWD=ZTD-ZHD

*ZWD* is usually obtained by non-difference and double-difference modes. This paper uses the Precise Point Positioning (PPP) software PRIDE PPP-AR(V 3.1) released by Wuhan University to calculate the absolute tropospheric delay of the site [16,17], and then the *PWV* is determined by the conversion factor using the following formula:(3)PWV=π×ZWD
where π denotes the water vapor conversion coefficient, which can be expressed as follows [18]:(4)$$\pi =\frac{10^{6}}{\rho_{W}R_{V}\left(k_{2}^{\prime }+\frac{k_{3}}{T_{m}}\right)}$$
where *ρ_W_* represents the density of water, and the value is 1000 kg/m^3^; *Rv* represents the specific gas constant of water vapor, which is 461.495 (J/kg^−1^ K^−1^). $k_{2}^{\prime }$ and *k*_3_ are atmospheric refractive index constants, and are 22.13 ± 2.2 (K/hPa) and (3.739 ± 0.012) × 10^−5^ (K^2^/hPa), respectively.

### 2.2. PWV Calculation by Integral Method

The accuracy of the *PWV* calculated by ERA5 reanalysis data is usually at the millimeter level; therefore, the *PWV* calculated by ERA5 data was used as a reference value for subsequent modeling and verification; the method of calculating *PWV* based on ERA5 reanalysis data is as follows [19,20]:(5)$$\mathrm{PWV}=\Sigma_{i=1}^{n}(\frac{q_{i}\times ∆p}{g})\;$$
where *q_i_* represents the specific humidity of the current pressure layer, ∆p represents the pressure difference between the upper and lower layers, and *g* is gravity acceleration, which is 9.80665 m/s^2^. The specific humidity of each layer is calculated by pressure *p* and water vapor pressure *p_w_* as follows [21]:(6)$$q=\frac{0.622\;\times \;P_{w}}{P-0.378\;\times \;p_{w}}$$
where the water vapor pressure *p_w_* is calculated using the following equation [22,23]:(7)$$p_{w}=\;(\frac{\mathrm{RH}}{100})\;\times \;(6.122\;\times \;\exp(\frac{17.67\;\times \;(T-273.15)}{\left(T-273.15\right)+243.5}))$$
where *RH* is the relative humidity of each layer, and *T* is the Kelvin temperature (K) of each layer. Through the above steps, the *PWV* of each grid can be calculated, and then the *PWV* of the station can be interpolated by the inverse distance weighting method and the grid data.

### 2.3. Establishment of a New Model for Water Vapor Inversion Based on Deep Learning

Random forest (RF) is a data mining technology developed by Breiman. It is a machine learning technique for classification and regression analysis. The basic principle of RF is to improve the decision tree algorithm; compared with traditional statistical methods, RF has higher computational accuracy, superiority under the same operating conditions, and better fitting to nonlinear data. In addition, RF can also evaluate the importance and relationship between variables, which is a feature that algorithms such as neural networks and support vector machines do not have [24,25].

In order to improve the computational efficiency and accuracy of machine learning, a function model as a prior value input to the model is established in this paper, and then the efficient nonlinear residual fitting ability of machine learning is used to improve the calculation accuracy of the model. In addition, taking into account the correlation between the period factor of the *PWV*, geographical factor, and zenith tropospheric delay, a new water vapor inversion model is established as follows:(8)$$\left\{\begin{array}{c} PWV=A\times ZTD+{\mathrm{Position}}_{\left(\mathrm{lat},\mathrm{lon},h\right)}+{\mathrm{Periodic}}_{\left(\mathrm{doy},\mathrm{hod}\right)}\\ Position=B_{1}\times lat+B_{2}\times lon+B_{3}\times h\\ Periodic=A_{0}+A_{1}\times \cos\left(2\pi \frac{\mathrm{doy}}{365.25}\right)+A_{2}\times \sin\left(2\pi \frac{\mathrm{doy}}{365.25}\right)+\\ A_{3}\times \cos\left(4\pi \frac{\mathrm{doy}}{365.25}\right)+A_{1}\times cos(4\pi \frac{\mathrm{doy}}{365.25})+A_{4}\times sin(4\pi \frac{\mathrm{doy}}{365.25})\\ +A_{5}\times cos(2\pi \frac{\mathrm{hod}}{24})+A_{6}\times sin(2\pi \frac{\mathrm{hod}}{24}) \end{array}\right.$$
where *A*, *A*_0_, *A*_1_, *A*_2_, *A*_3_, *A*_4_, *A*_5_, *A*_6_, *B*_1_, *B*_2_, *B*_3_ are coefficients; *ZTD* is the total delay of the zenith troposphere, lat, lon, h are latitude, longitude, and geodetic height of the station, respectively; doy is day of year; hod is the hourly time of day.

Selecting the optimal parameter combination is very important for the machine learning model, as the optimal parameter combination can not only improve the accuracy of the calculation results but also improve the calculation speed of the model. The grid search method is used to determine the optimal parameter combination in this paper. The overall process is shown in Figure 1, and the steps for establishing the model are as follows.

Step 1: the ERA5_PWV is calculated by using the multi-layer pressure data provided by ERA5, and the function model is constructed by using the ERA5_PWV and the ZTD calculated by GNSS as a priori value, which is recorded using the least squares model (LSQ Model). In this paper, a prior value is calculated and input into the random forest model to predict the PWV value, which is equivalent to using the random forest model to calculate the difference between the LSQ model and the reference value. The advantage of this method is it reduces the dependence on the intelligent model, only making full use of the fitting ability of machine learning for nonlinear residuals, thus not only improving the accuracy of the model but also reducing the running time of the model.

Step 2: the model is trained by using the calculated prior value, geographical, time and ZTD as input parameters. The data of the years 2021 and 2022 are used as the training set, and the data of the year 2023 are used as the verification set. The grid search method is used to configure the optimal parameters of the model to prevent over-fitting.

Step 3: after training the RF model according to the constraints and training data, the PWV calculated by the random forest model is recorded as RF_PWV, the PWV calculated by the traditional method using Tm is recorded as GNSS_PWV, and the PWV value calculated by the radiosonde data is recorded as RS_PWV. Finally, the accuracy information of GNSS_PWV, RS_PWV, and RF_PWV are compared and analyzed.

### 2.4. Datasets

The dataset of this experiment mainly includes ERA5 data, GNSS data, radiosonde data, and GSMaP data products.

#### 2.4.1. ERA5 Data

The ERA5 reanalysis data were released in July 2017; they are the fifth generation of global climate products produced by ECMWF and a major upgrade to the previous generation of ERA-Interim datasets. They have been widely used in meteorological and hydrological research. In this paper, the ERA5 data mainly include two forms. One uses the multi-layer pressure data to calculate the PWV as the reference value for water vapor validation [26]. The other compares and analyzes the spatial and temporal variations in water vapor based on the rainfall data provided by ERA5 [27].

#### 2.4.2. GNSS Data

In this paper, the GNSS data used are from the Hong Kong CORS network. The data of 8 stations were selected as the training sets, and the data of 9 stations were selected as the verification set; the detailed station selection and distribution are shown in Figure 2.

#### 2.4.3. Radiosonde Data

In order to verify the effectiveness of the new model, the radiosonde data on Hong Kong were used in this paper, and can be downloaded from University of Wyoming Atmospheric Science Radiosonde Archive (Available online: https://weather.uwyo.edu/upperair/bufrraob.shtml (accessed on 1 August 2024)). Radiosonde data contain atmospheric data from the surface to the top of the atmosphere, and there are at least 30 pressure levels at UTC 00:00 and 12:00 every day [28,29]. Due to the radiosonde data being densely distributed on the vertical section and having high accuracy, the PWV obtained from radiosonde data is often used as a reference value. The spatial distribution of the radiosonde station is also shown in Figure 2.

#### 2.4.4. GSMaP Data

GSMaP (Global Satellite Mapping of Precipitation) is a global precipitation monitoring program led by the Japan Meteorological Agency (JMA, Tokyo, Japan), which aims to provide accurate and comprehensive global precipitation data through satellite remote sensing technology. Since its launch in 2002, GSMaP data have become an important data source in the fields of meteorology, climatology, environmental science, water resources management, and so on. The system is based on satellite observations and combined with ground observation station data, using advanced algorithms for the real-time monitoring and analysis of global precipitation, covering a global scope and providing precipitation information with high spatial and temporal resolution [30]. These data can be downloaded from the GSMaP data website (available online: https://sharaku.eorc.jaxa.jp/GSMaP/index.htm (accessed on 27 August 2024)). The data used in this study were obtained from GSMaP_Gauge with a spatial-temporal resolution of 1 h/0.01°.

## 3. Experimental Results and Analysis

In the experiment, the PWV obtained by ERA5 data was used to establish the model and verify the feasibility of the new model. Meanwhile, the reliability of the model was verified by radiosonde data in the experimental area. The specific schemes were as follows. Scheme 1: the ERA5_PWV was used as the reference value to establish the model; the accuracy of the PWV obtained between the traditional method (GNSS_PWV), the prior model (P_PWV), and the new model (RF_PWV) proposed in this paper was analyzed to verify its feasibility. Scheme 2: taking the PWV calculated by the radiosonde data as the reference value, the accuracy evaluation and residual analysis of the PWV calculated by the P_Model and RF_PWV were carried out to verify its reliability; finally, it was compared with the results of the LSTM model, and the advantages of the new model were verified.

### 3.1. Feasibility and Accuracy Analysis of the RF_PWV Model

The results of the PWV accuracy information calculated according to Scheme 1 are shown in Figure 3, and it can be seen that the accuracy of the RF_PWV model has been significantly improved. The BIAS, MAE, and RMSE values of different stations are relatively stable, such as the BIAS value falling within 0.2 mm, the MAE value being about 0.8 mm, and the RMSE value being about 1 mm. At the same time, after the new model (RF_PWV) was processed by machine learning, the PWV retrieved by the RF_PWV was greatly improved in each accuracy index, which proves that machine learning has strong nonlinear fitting ability. Compared with the traditional method (GNSS_PWV), the new model not only has a simple model structure, but also reduces the observational and computational errors introduced by calculating the weighted mean temperature from measured meteorological parameters.

In order to further verify the accuracy of the RF_PWV model, the average value of each precision index and the maximum value of residual error of 9 GNSS stations were calculated, and the results are shown in Table 1. It can be seen that the traditional calculation method can meet the accuracy requirements of water vapor inversion, but the maximum deviation value reaches 11.2 mm while the maximum deviation value of the PWV calculated by the RF_PWV model is only 6.2 mm, which indicates that the RF_PWV model gives full play to the nonlinear residual fitting ability of the intelligent algorithm, calculates a more accurate peak, and shows that different accuracy indicators have been significantly improved, with an average increase of at least 38%.

### 3.2. Reliability Analysis of the RF_PWV Model

In order to verify the reliability of the model, based on the long-short term memory neural network model (LSTM), this paper used the same training dataset as Section 2.3, and used the sounding station data from the Hong Kong region in 2023 as the reference value. Compared with the random forest algorithm, LSTM has the “memory gate” function of recording the periodic changes in data, so it is often used to process time series data. Secondly, in order to better learn the changes in time series data, LSTM needs to set the size of the sliding window reasonably to predict data, which usually takes a lot of time to debug, while the RF model does not need to consider this problem. In this regard, this paper analyzes the residuals of the prior values of the function model, the PWV values calculated by the RF model and the LSTM model, and finally calculates the residual results of the ERA5 reanalysis data. The calculation results are shown in Figure 4. It can be seen that the calculation results of the function model conform to the normal distribution, but there is a certain system error, which causes the fitted curve to shift to the right. After calculating and processing the prior values using the random forest algorithm, it can be seen that the overall residual of the model accounts for a larger proportion near 0, which has a significant improvement effect. At the same time, the LSTM model also shows the same improvement effect. However, due to the limitation of the sliding window step size and the prediction step size, a certain error accumulation is caused, resulting in an increase in the proportion of some larger residuals compared with the RF model. This shows that the RF model can provide high-precision results, and the use of the RF model is simpler than the LSTM model. Finally, the residual results of the ERA 5 reanalysis data are calculated, and it is found that the error with the sounding station results is very small, which is consistent with the calculation results of many scholars, and once again proves the reliability of the model. In order to further evaluate the accuracy of the model, this paper calculate d accuracy evaluation factors of the three models. As shown in Table 2, it can be seen that there was a significant improvement after using the RF model and the LSTM model. Each accuracy index was significantly reduced, and the values were about 1 mm.

## 4. Analysis of Water Vapor Inversion and Temporal and Spatial Variation Characteristics of Rainstorm

Affected by Typhoon Haikui, heavy rain occurred in Hong Kong from 6:00 to 12:00 on 7 September 2023, and rainfall values of more than 70 mm were reported and recorded. In order to better evaluate the spatial and temporal characteristics of the PWV retrieved by the new model (RF_PWV) proposed in this paper and analyze its response relationship with rainstorm events, the rainfall products provided by ERA5 and GSMaP were selected as a reference for comparative analysis.

### 4.1. Water Vapor Inversion Based on the RF_PWV Model and Its Time Series Analysis

Based on the RF_PWV model, the water vapor information of nine GNSS stations with a day of year 246 to 262 (UTC time of 3–19 September 2023) for 17 consecutive days was obtained, and the PWV time series is shown in Figure 5. It can be seen that after the residual pressure of Typhoon Haikui began to land in Hong Kong, the PWV values of each station showed a significant upward trend, and rainfall events occurred after the PWV reached its peak; once the residual typhoon was far away from Hong Kong, the PWV of each station began to decrease gradually. It is worth noting that the PWV of the KYC1 station was significantly lower than that of other stations, which is consistent with the results of water vapor accuracy index in Figure 3, as the multipath error causes the tropospheric delay parameters calculated by the station to be greatly affected. In order to verify this conclusion, the multipath error information of KYC1 was calculated in this paper, at the same time, the inland stations were selected to calculate the multipath error information separately. The multipath error results of the six stations are shown in Figure 6, and it was found that the multipath errors of HKKT, HKQT, HKCL, T430, and HKTK stations were significantly smaller than that of KYC1 near the sea.

In order to analyze the PWV changes before and after the rainstorm event in detail, the PWV time series of the HKCL, HKKT, and T430 stations were compared and analyzed by using ERA5 and GSMaP rainfall products, and the results are shown in Figure. 7. It can be seen that there are differences in the rainfall products provided by ERA5 and GSMaP, and the rainfall estimation obtained by GSMaP is significantly lower than that obtained by ERA5. It is possible that underestimation of the GSMaP rainfall product occurred, which leads to the result missing many smaller rainfall events. On the contrary, the rainfall results obtained by ERA5 were overestimated. In summary, the two rainfall products are able to provide a good response to the heavy rainstorm event with a day of year of 250 (the UTC time of 7 September 2023), and the maximum value of two rainfall products reaches about 14 mm per hour. The PWV information inverted by the RF_PWV model can respond well to the occurrence of rainfall events. It continues to rise before rainfall, and the peak value of the PWV is close to 74 mm, and after the maximum rainfall, it decreases with the end of rainfall, and the PWV decreases to the minimum. The variation information on the increase and decrease in the PWV values before and after the rainfall at the three stations was further analyzed, as shown in Table 3, Table 4, and Figure 7. It can be seen that the PWV series of the three stations increased by up to 12 h on the day of year of 248 (the UTC time of 5 September 2023). Then, a rainfall event occurred after a small decrease, as shown in the black circle labels in Figure 7, and it was verified that the aggregation of clouds and the increase in humidity before rainfall led to the increase in the PWV, and there was a certain lag between the increase in the PWV value and the occurrence of rainfall events. After continuous rainfall, the water vapor in the air was continuously taken away by the rainfall, resulting in a continuous decline in the PWV time series, as shown in the blue circle labels in Figure 7. Then, the PWV rises slightly because the surface precipitation forms a part of water vapor after evaporation but the temperature is not high after rainfall, causing the PWV series to show a small increase.

In summary, there is a strong correlation between the PWV series calculated by the RF_PWV model and rainfall, and rainfall occurs easily after the saturation of water vapor. Not all the peak values of the PWV must correspond to the occurrence of rainfall events, but an increase in water vapor is one of the necessary prerequisites for rainfall [31]. When the rainfall begins to weaken, the PWV decreases sharply, which continues until the end of the rainfall. When the consumed water vapor is replenished, the PWV time series gradually rises and finally tends to be stable.

### 4.2. The Spatial and Temporal Characteristics of PWV Inverted by the RF_PWV Model

In order to analyze the spatial and temporal variation in the PWV before and after heavy rain, based on ERA5 and GSMaP rainfall data products, the spatial interpolation method was used to obtain the rainfall information with, a time interval of 12 h. Figure 8 shows the spatial distribution result of rainfall obtained by ERA5, and Figure 9 shows the spatial distribution result of rainfall obtained by GSMaP. It can be seen that although there are local differences in the spatial distribution of the two rainfall results, the overall trend is basically the same, and both rainfall events occurred on 7 and 8 September.

In addition, based on the RF_PWV model, the spatial distribution of the PWV was obtained every 3 h. The results are shown in Figure 10, and it can be seen that the water vapor at the HKKT and T430 stations in the north of Hong Kong continued to rise from 6:00 to 12:00 on 7 September, and reached 74 mm at about the time of 12:00. Subsequently, the water vapor decreased briefly from 12:00 to 15:00, then continued to rise and accumulate to 74 mm from 15:00 to 24:00, and then began to decrease. The spatial and temporal variation in these whole PWV values are consistent with the spatial and temporal variation in rainfall shown in Figure 8 and Figure 9. The rainfall information obtained by GSMaP as shown in Figure 9, is consistent with the PWV peak value results calculated by the RF_PWV model. The first rainfall event was concentrated near the HKKT and T430 stations, and the second maximum rainfall occurred near the HKCL and HKQT stations, which proves that a reliable spatial and temporal distribution of water vapor can be obtained by using the RF_PWV model. It is worth noting the variation in Figure 10c,g, and it can be seen that the PWV peak over the whole Hong Kong area has a tendency to shift southward, which is consistent with the spatial distribution of rainfall in shown Figure 9a,b. The maximum value of rainfall in Figure 9a appears at the HKTK station in the northern part of Hong Kong, and the maximum value of rainfall in Figure 9b appears at HKCL and HKQT in the southern part of Hong Kong, which is consistent with the distribution of the PWV peak shown in Figure 10g, further illustrating the accuracy of the RF_PWV model in water vapor inversion. It also proves that GSMaP rainfall data are more applicable in small areas, with high temporal resolution. In addition, the temporal and spatial variation information of the PWV with a time resolution of 3 h were obtained by using ERA5 data. The results are shown in Figure 11 and were further verified and analyzed with the water vapor inverted by the above RF_PWV model. It can be seen that the water vapor results calculated by the RF_PWV model are consistent with those obtained by ERA5 in terms of temporal and spatial variations, and the PWV of the HKKT and T430 stations in northern Hong Kong continued to accumulate from 6:00 to 12:00 on September 7. The PWV reached its peak at the time of 24:00 and then decreased again, finally continuing to rise again. These variations are consistent with the calculation results of the RF_PWV model, which shows that the water vapor distribution calculated by the ERA5 and RF_PWV models is different at the KYC1 station, but the overall trend of its spatial and temporal variations is consistent. The main reason for this phenomenon is that the KYC1 station is adjacent to the sea area. The ZTD calculated by the GNSS data of the KYC1 is greatly affected by the multipath error, which indirectly affects the PWV inversion results, and this is consistent with the analysis conclusion in Section 4.1. However, under the premise of satisfying the timeliness, the ability to calculate water vapor based on the RF_PWV model is considerable.

## 5. Conclusions

In this paper, based on a machine learning algorithm, a new PWV inversion model without weighted average temperature was established by using two years of meteorological data and GNSS data. The new model was used to analyze the temporal and spatial changes in rainstorm events in Hong Kong. The following conclusions were obtained through experimental analysis:(1)When ERA5 data are used as the reference value, the water vapor accuracy indexes BIAS, MAE, and RMSE of the new model RF_PWV inversion proposed in this paper are 0.2 mm, 0.8 mm and 1.1 mm, respectively. Compared with the traditional method, the inversion accuracy index value of RF_PWV is increased by 38% on average. When the sounding station data are used as the reference value, the accuracies of the new model and the LSTM model are equivalent, and the accuracy index is within 1 mm. However, under the same conditions, the RF model has a simple structure and is easy to implement, which can effectively avoid the error accumulation caused by the algorithm.(2)In response to the rainstorm event caused by Typhoon Haikui, the ERA5 and GSMaP rainfall products are used to analyze the water vapor time series inverted by the RF_PWV model. The results show that with the residual typhoon landing in Hong Kong, the PWV time series obtained by the RF_PWV model shows a continuous upward trend before the rainfall event, and then the PWV time series gradually decreases after the rainfall event. The whole PWV time series can better respond to the rainstorm event.(3)By analyzing the spatial and temporal distribution results of the PWV retrieved by the RF_PWV model, it can be seen that with the occurrence of rainstorm events, the spatial and temporal variations in the PWV show a corresponding rising and falling phenomenon, and when the actual rainfall reaches the peak, the spatio-temporal information of the PWV also corresponds to the peak value of the whole region. At the same time, the spatio-temporal distribution of the PWV inverted by the RF_PWV model is consistent with the trend in water vapor obtained by ERA5, which verifies the reliability of the water vapor retrieved from the RF_PWV model.

## Figures and Tables

**Figure 1 sensors-25-00420-f001:**
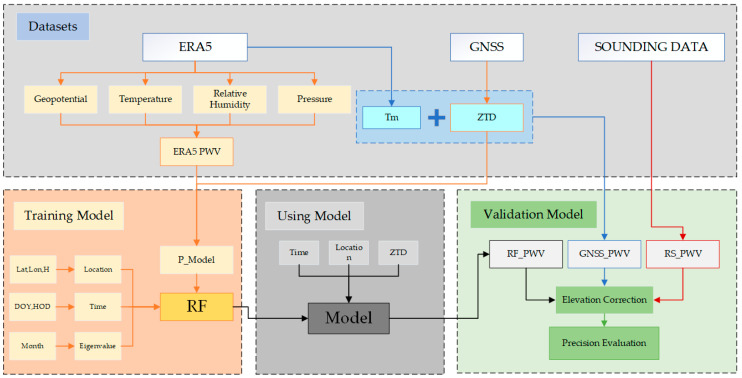
Flow char of RF_PWV model.

**Figure 2 sensors-25-00420-f002:**
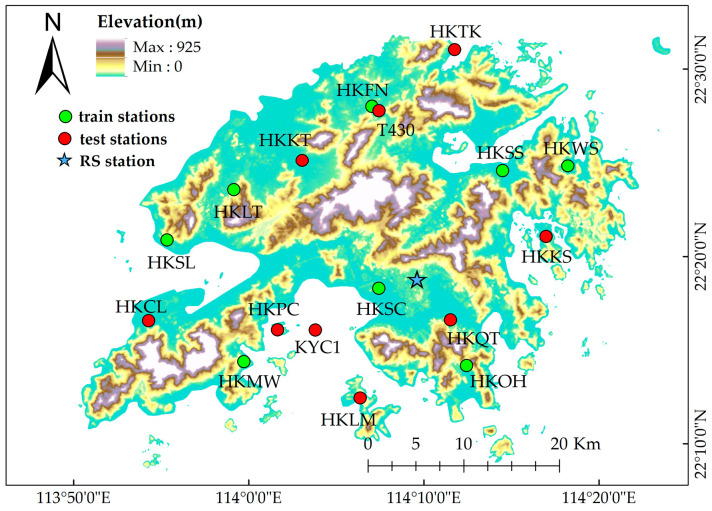
Distribution of GNSS stations and the radiosonde station.

**Figure 3 sensors-25-00420-f003:**
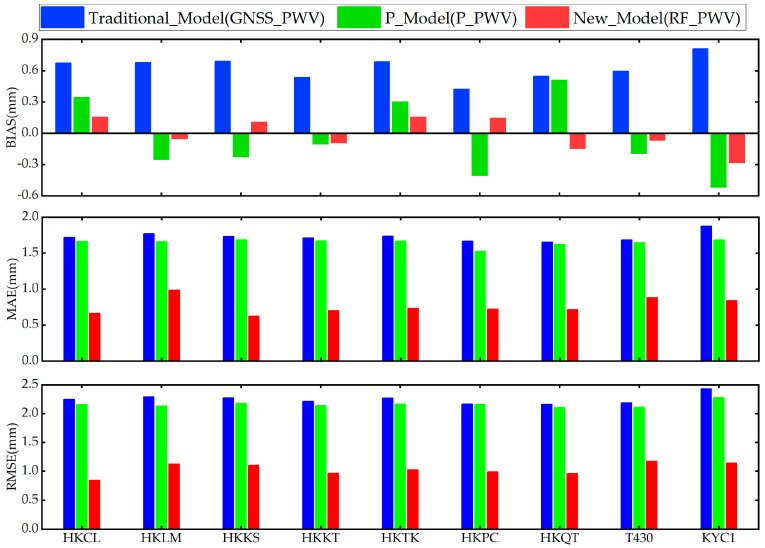
The accuracy index results of 9 GNSS stations are calculated by different water vapor models.

**Figure 4 sensors-25-00420-f004:**
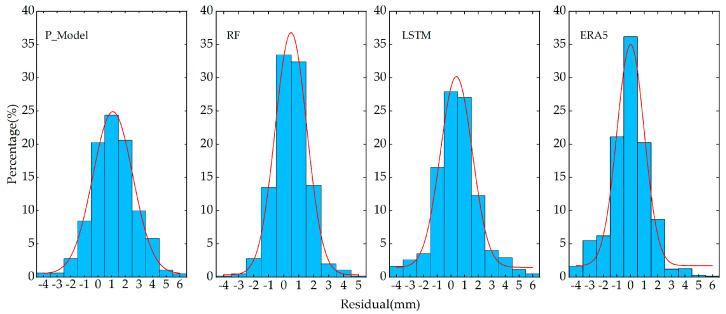
The percentage of water vapor residuals obtained by the three models, respectively, and ERA 5.

**Figure 5 sensors-25-00420-f005:**
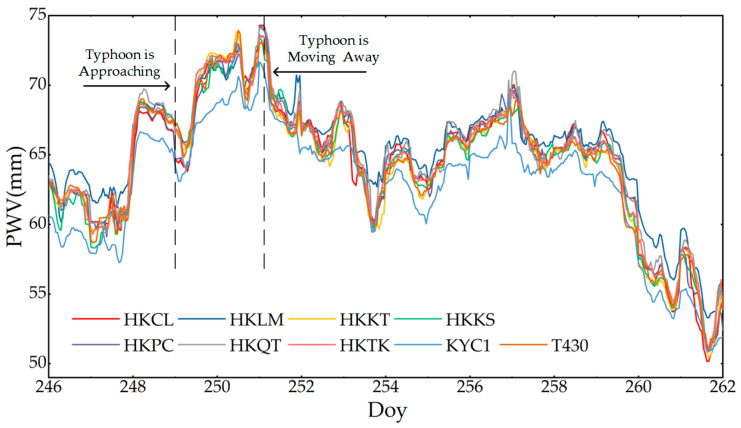
Response results of PWV and rainfall events at different stations.

**Figure 6 sensors-25-00420-f006:**
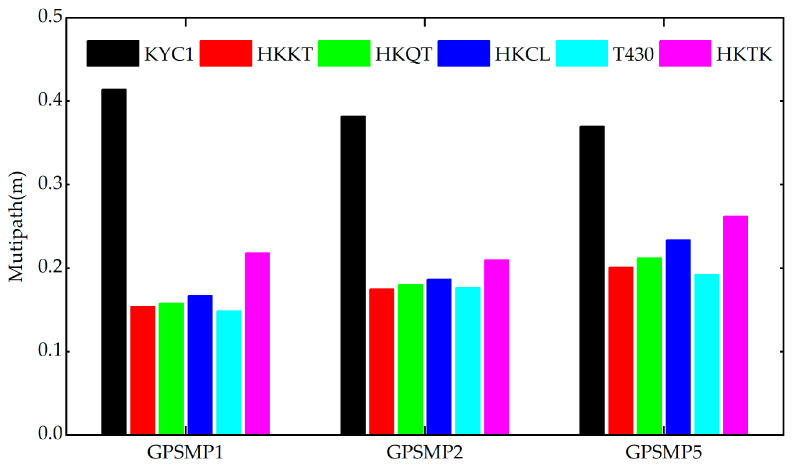
Multipath error results of different stations.

**Figure 7 sensors-25-00420-f007:**
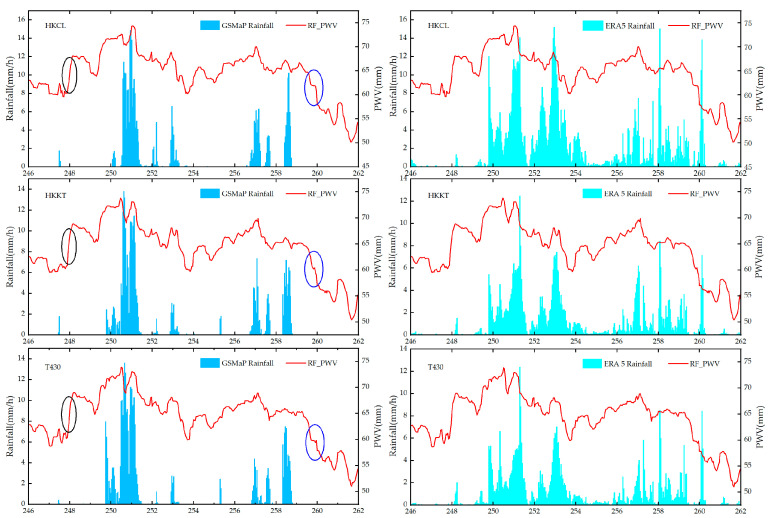
PWV time series inverted by RF_PWV model.

**Figure 8 sensors-25-00420-f008:**
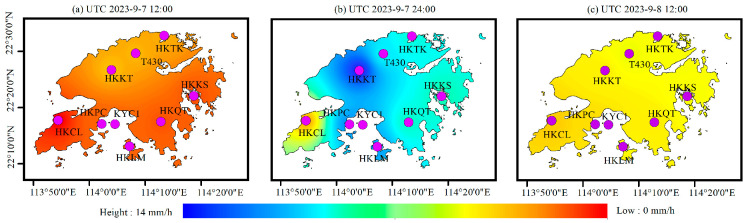
The spatial distribution of rainfall information based on ERA5.

**Figure 9 sensors-25-00420-f009:**
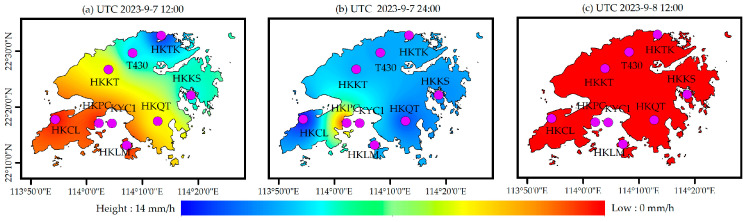
The spatial distribution of rainfall information based on GSMaP.

**Figure 10 sensors-25-00420-f010:**
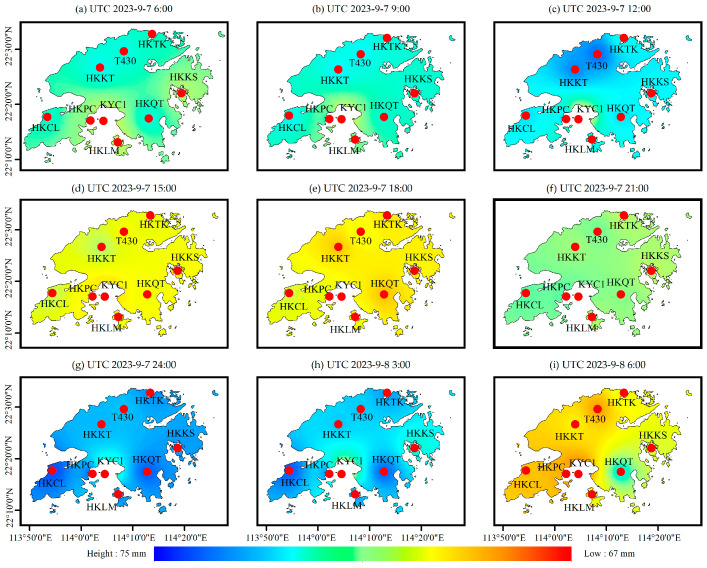
The spatial and temporal variation in PWV based on RF_PWV model.

**Figure 11 sensors-25-00420-f011:**
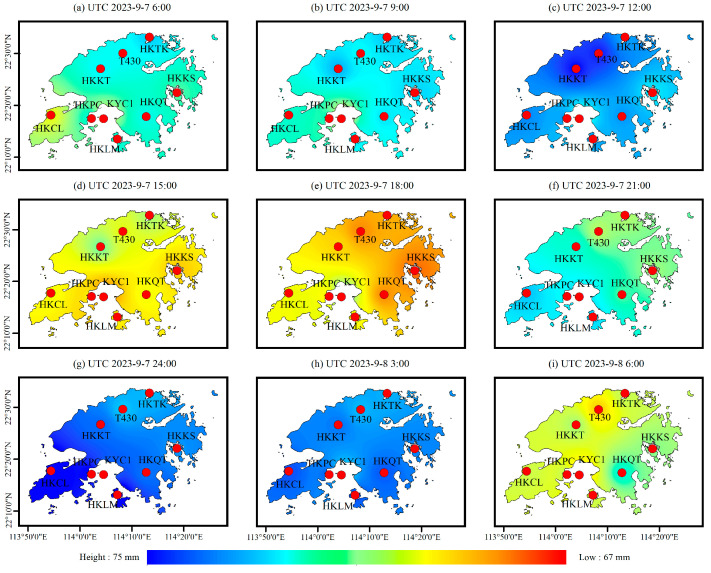
The spatial and temporal variation in PWV based on ERA 5.

**Table 1 sensors-25-00420-t001:** The accuracy index and residual maximum results of different models (mm).

Value	GNSS_PWV	RF_PWV
Max Residual	11.2	6.2
BIAS	0.6	0.2
MAE	1.7	0.8
RMSE	2.2	1.1

**Table 2 sensors-25-00420-t002:** Evaluation of water vapor accuracy information obtained by three models (mm).

Value	P_PWV	RF_PWV	LSTM_Model
BIAS	1.5	0.8	0.8
MAE	1.6	1.0	1.1
RMSE	1.7	1.2	1.2

**Table 3 sensors-25-00420-t003:** The variation information of PWV before rainfall events (black circle labels in Figure 7).

Rainfall Events at Corresponding GNSS Station	PWV (mm)			Interval (h)	Rate of Variation (mm/h)
	Max	Min	Variation Value		
HKCL	68.1	59.6	8.5	12	0.7
HKKT	68.9	60.4	8.5	11	0.8
T430	69.1	59.6	9.5	12	0.8

**Table 4 sensors-25-00420-t004:** The variation information of PWV after rainfall events (blue circle labels in Figure 7).

Rainfall Events at Corresponding GNSS Station	PWV (mm)			Interval (h)	Rate of Variation (mm/h)
	Max	Min	Variation Value		
HKCL	66.2	56.0	10.2	16	0.6
HKKT	65.2	56.9	8.3	16	0.5
T430	65.3	55.7	9.6	16	0.6

## Data Availability

The raw data supporting the conclusions of this article will be made available by the authors on request.

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
