# Peer review of "Establishment and Evaluation of Atmospheric Water Vapor Inversion Model Without Meteorological Parameters Based on Machine Learning"

_sensors, 2025, doi:10.3390/s25020420_

Round 1
Reviewer 1 Report
Comments and Suggestions for Authors
Utilizing the Random Forest algorithm, the manuscript introduces an innovative GNSS Precipitable Water Vapor (PWV) conversion model that eschews the need for meteorological parameters. This model stands out for its universality, achieving accuracy on par with the Long Short-Term Memory (LSTM) neural network model. Moreover, the model was deployed to compute and dissect the alterations in the spatial distribution of water vapor during the Haikui typhoon incident in Hong Kong in 2023. The resultant experimental data confirmed the model’s viability in the inversion of water vapor and the delineation of its spatial distribution shifts. While the model and its technical methodology are lucidly detailed, there remain certain issues that warrant a more comprehensive elucidation:
1. The author analyzed the feasibility of the new model through examples, what is the reliability of the new model in GNSS PWV inversion?
2. Can the new GNSS PWV inversion model proposed in the paper be used for real-time or near real-time GNSS water vapor inversion?
3. In Section 4.2 of the paper, in order to analyze the spatial distribution changes of water vapor during the Haikui typhoon event in Hong Kong, the author used the new model to obtain the precipitable water vapor distribution changes with a time resolution of 3 hours, and verified them using ERA5 reanalysis products, why the precipitable water vapor distribution changes obtained by the new model and ERA5 is numerically different.
The author 's revised version can be accepted for publication.

Author Response
Response to Reviewer 1:
Thanks for your comments on our paper. The following is the answers and revisions:
Comments 1: The author analyzed the feasibility of the new model through examples, what is the reliability of the new model in GNSS PWV inversion?
Response 1: Through Figure 3 in Section 3.1, it can be seen that the inversion accuracy of the new model of each station was better than that of the traditional water vapor conversion model, which proved the feasibility of the model. In Section 3.2, the data of sounding station were used as reference values, and the reliability of the model was also verified by comparing the residual distribution of ERA5 data and the new model. Finally, Figure 10 and 11 in Section 4.2 showed that the spatial and temporal variation of PWV inverted by the new model was consistent with the spatial and temporal variation of ERA5 PWV, which also proved the reliability of the article.
Comments 2: Can the new GNSS PWV inversion model proposed in the paper be used for real-time or near real-time GNSS water vapor inversion?
Response 2: Based on the GNSS water vapor inversion model proposed by machine learning, it can be seen from Figure 1 in Section 2.3 that the input parameters of the model only need to measure the longitude, latitude, elevation and ZTD of the station, so real-time or near real-time ZTD values were provided. The new model proposed in this paper can be used for real-time or near real-time water vapor inversion.
Comments 3: In Section 4.2 of the paper, in order to analyze the spatial distribution changes of water vapor during the Haikui typhoon event in Hong Kong, the author used the new model to obtain the precipitable water vapor distribution changes with a time resolution of 3 hours, and verified them using ERA5 reanalysis products, why the precipitable water vapor distribution changes obtained by the new model and ERA5 is numerically different.
Response 3: This small difference had a variety of error components. Firstly, the new model in this paper used ZTD to directly convert into PWV, and its accuracy was better than the traditional GNSS-PWV model. Therefore, the calculation results of the new model were reliable. However, in order to unify with the planar data provided by ERA 5, this paper used the spatial interpolation method to make the point data calculated by the new model into planar data, which may be the reason for the slight difference.

Reviewer 2 Report
Comments and Suggestions for Authors
The paper presents a method for converting zenith total delay (ZTD) estimated by GNSS to precipitable water vapor (PWV). The method is based on the machine learning algorithm random forest. I think the results seem to be rather good and impressive. However, there are several parts which are unclear and needs to be improved.
From what I understood about how the proposed algorithm work, first an a priori PWV is estimated from the ZTD based on equation 7 (P_model). Then a correction to this is obtained from the random forest method. Given that the input data seems to be only the ZTD, the position and the time, I think the results are surprisingly good. Normally I would think that it would be much better to first use the surface pressure to estimate the ZHD, then reducing this from the ZTD to get the ZWD, which is then used to get the PWV. The reason is that I would assume that the atmospheric pressure (hence the ZHD) could not be predicted well using just the location and the time. But maybe the pressure is easy to predict in Hong Kong. I still wonder if the results would be even better if the ZWD was used as input to the algorithm instead of ZTD.
It is not clear to me what the “traditional method” used in this paper really is. There exist many different models for Tm or the conversion factor between ZWD and PWV, which one were used? The commonly used Bevis model, or a model more optimized for the region? What was the input parameters of this model and how were these obtained?
In section 3.2, a comparison is also made to the LSTM (long-short term memory). The description of this method seems to be missing.
A bit more details on the GNSS processing done to obtain the ZTD could be described in more detail.
Line 54-56: “Tm usually needs to be calculated by the measured temperature, pressure data of the station, which leads to the limitation of the timeliness of PWV inversion”. Why does this lead to limitations in the timeliness? it is not clear.
Line 99-101: “Compared with the double-differenced mode, the un-differenced mode can obtain the absolute tropospheric delay of stations.”. It is true that the absolute ZTD is difficult to estimate using double differencing for a small network, however, for larger networks this is not an issue.
Line 157-158: “Compared with the model that only predicts PWV based on time and geographical location,” It is unclear what model is referred to here.
Figure 1: For estimating the GNSS_PWV, I guess that the surface pressure (or the ZHD) is also needed, in addition to the ZWD and Tm.
Line 233-237: “Compared with the traditional method(GNSS_PWV), the RF_PWV model does not need the weighted mean temperature, which not only has a simple model structure, but also reduces the observational and computational errors introduced by calculating the weighted mean temperature from measured meteorological parameters.” There exist other methods for calculating the conversion factor between ZWD and PWV which do not model Tm explicitly.
Figure 4: The result from P-model, RF, and LSTM are compared to radiosondes. Maybe it would be interesting to also add ERA5.
Line 287-295: it is discussed that the KYC1 station have lower PWV than other stations. The text refers to Figure 5, but KYC1 is not shown in this figure. However, in fig 7 it is clear that KYC1 is several mm lower than other stations. It is said that this is consistent with fig 3, however, in fig 3 the bias of KYC1 is only -0.3 mm, not several mm. Do the problems with KYC1 vary in time? The authors argue that the issue with KYC1 is problems with multipath. This might very well be the case. Are there anything in the surrounding of KYC1 (reflecting objects etc.) which could cause significant multipath. If there are clear evidence of significant multipath at this station, maybe it should be excluded from the analysis.
Figures 10 and 11: Use a different color map. Right now, reddish colors are both indicating high PWV and rather low PWV. Also, try using the same scale on the colorbar for the two figures (right now there is only a 0.4 mm difference, seems unnecessary).
Figure 11: Should it be ERAY PWV instead of RF PWV (the text seems to indicate this, although it is not completely clear). If not, and it is indeed the RF PWV shown in the figure, what is the difference to figure 10?
Line 441-442: The names of this reference are wrong, the first and last names seems to have been mixed. It should be “Elgered, G; Johansson, J. M., Rönnäng, B. O.”. Check the other references for similar mistakes.
Comments on the Quality of English LanguageIn general, I think the paper is in need of proof-reading and English language of the paper needs to be improved. The best would be to have I checked by a native speaker. Right now, the paper is sometimes difficult to understand. On several places there are some strange repetitions, like “As can be seen from Figure 5 It can be seen from Figure 5” (line 282-283) and “The results are shown in Figure 11, which is further verified and analyzed with the water vapor inverted by the above RF _ PWV model, the results are shown in Figure 11, which is further verified and analyzed with the water vapor inverted by the RF_PWV model.” (line 375-378), indicating that the paper has not been proof-read very carefully.
Author Response
Comments 1: From what I understood about how the proposed algorithm work, first an a priori PWV is estimated from the ZTD based on equation 7 (P_model). Then a correction to this is obtained from the random forest method. Given that the input data seems to be only the ZTD, the position and the time, I think the results are surprisingly good. Normally I would think that it would be much better to first use the surface pressure to estimate the ZHD, then reducing this from the ZTD to get the ZWD, which is then used to get the PWV. The reason is that I would assume that the atmospheric pressure (hence the ZHD) could not be predicted well using just the location and the time. But maybe the pressure is easy to predict in Hong Kong. I still wonder if the results would be even better if the ZWD was used as input to the algorithm instead of ZTD.
Response 1: The traditional GNSS-PWV inversion model was described by the reviewer. Firstly, the ZHD was calculated, and then the ZWD was calculated by using ZTD minus ZHD. Finally, the PWV value was calculated by using the conversion coefficient between ZWD and PWV. The steps were as follows : Eq.(1)~Eq.(3). We recognized the reviewer 's point of view, but the original intention of our design experiment was to propose a conversion model between ZTD and PWV without meteorological parameters, so we did not consider using ZWD as an input parameter. However, in order to prove your point of view, we used ZWD as a parameter to calculate the data of three stations, and the results were shown in Fig 1.
Comments 2: It is not clear to me what the “traditional method” used in this paper really is. There exist many different models for Tm or the conversion factor between ZWD and PWV, which one were used? The commonly used Bevis model, or a model more optimized for the region? What was the input parameters of this model and how were these obtained?
Response 2: It is not clear to me what the “traditional method” used in this paper really is. There exist many different models for Tm or the conversion factor between ZWD and PWV, which one were used? The commonly used Bevis model, or a model more optimized for the region? What was the input parameters of this model and how were these obtained?
Comments 3: In section 3.2, a comparison is also made to the LSTM (long-short term memory). The description of this method seems to be missing.
Response 3: In accordance with your requirements, we had added a detailed description of the LSTM model in Section 3.2 of this article.
Comments 4: A bit more details on the GNSS processing done to obtain the ZTD could be described in more detail.
Response 4: According to your requirements, the process of GNSS solving ZTD has been described in more detail in Section 2.1 of this article.
Comments 5: Line 54-56: “Tm usually needs to be calculated by the measured temperature, pressure data of the station, which leads to the limitation of the timeliness of PWV inversion”. Why does this lead to limitations in the timeliness? it is not clear.
Response 5: Because most GNSS monitoring stations did not have the observation of meteorological parameters, they can only provide non-real-time meteorological parameters of nearby stations or atmospheric analysis data, which limits the timeliness of GNSS PWV inversion.
Comments 6: Line 99-101: “Compared with the double-differenced mode, the un-differenced mode can obtain the absolute tropospheric delay of stations.”. It is true that the absolute ZTD is difficult to estimate using double differencing for a small network, however, for larger networks this is not an issue.
Response 6: Thank you very much for the reviewer 's comments, we had revised the corresponding description of lines 107-110.
Comments 7: Line 157-158: “Compared with the model that only predicts PWV based on time and geographical location,” It is unclear what model is referred to here.
Response 7: In the inversion process of GNSS_PWV, not only ZWD and Tm, but also ground pressure is required. In the process of calculating ZTD in PPP mode, the Saastamoinen model was used to estimate the ground pressure value. This paper was not shown in Figure 1, because this paper mainly wanted to propose a conversion model between GNSS ZTD and PWV without meteorological parameters. In order to make the flow chart more intuitive, only ZTD was written in the GNSS-PWV inversion process.
Comments 8: Figure 1: For estimating the GNSS_PWV, I guess that the surface pressure (or the ZHD) is also needed, in addition to the ZWD and Tm.
Response 8: The previous expression was inappropriate and gave you a misunderstanding, we have revised it.
Comments 9: Line 233-237: “Compared with the traditional method(GNSS_PWV), the RF_PWV model does not need the weighted mean temperature, which not only has a simple model structure, but also reduces the observational and computational errors introduced by calculating the weighted mean temperature from measured meteorological parameters.” There exist other methods for calculating the conversion factor between ZWD and PWV which do not model Tm explicitly.
Response 9: Many scholars had established different Tm models according to different regions to achieve the purpose of high-precision GNSS_PWV inversion. However, when using the conversion coefficient between ZWD and PWV for inversion, Tm was provided by ERA5 reanalysis data, so there's no modelling of Tm.
Comments 10: Figure 4: The result from P-model, RF, and LSTM are compared to radiosondes. Maybe it would be interesting to also add ERA5.
Response 10: Percentage residuals of PWV values calculated by ERA5 with sounding station data as reference values had been added in accordance with your recommendations.
Comments 11: Line 287-295: it is discussed that the KYC1 station have lower PWV than other stations. The text refers to Figure 5, but KYC1 is not shown in this figure. However, in fig 7 it is clear that KYC1 is several mm lower than other stations. It is said that this is consistent with fig 3, however, in fig 3 the bias of KYC1 is only -0.3 mm, not several mm. Do the problems with KYC1 vary in time? The authors argue that the issue with KYC1 is problems with multipath. This might very well be the case. Are there anything in the surrounding of KYC1 (reflecting objects etc.) which could cause significant multipath. If there are clear evidence of significant multipath at this station, maybe it should be excluded from the analysis.
Response 11: We had modified the correct image serial number. Firstly, the accuracy index of Figure 3 was calculated by using the RF_PWV results and reference values of a single station in 2023. Figure 7 was the comparison of RF_PWV results of different stations in 16 days. The reference values of the two figures were different, so there will be different magnitude errors. Secondly, KYC1 had serious multipath error effects, but in order to verify the spatial and temporal changes of the model, the calculation results of multiple stations were selected as much as possible.
Comments 12: Figures 10 and 11: Use a different color map. Right now, reddish colors are both indicating high PWV and rather low PWV. Also, try using the same scale on the colorbar for the two figures (right now there is only a 0.4 mm difference, seems unnecessary).
Response 12: According to your requirements, we had changed the colors of Figure 10 and Figure 11 and used the same scale axis.
Comments 13: Figure 11: Should it be ERAY PWV instead of RF PWV (the text seems to indicate this, although it is not completely clear). If not, and it is indeed the RF PWV shown in the figure, what is the difference to figure 10?
Response 13:Thanks to the reviewer 's reminder, we had changed the name of the picture in response.
Comments 14: Line 441-442: The names of this reference are wrong, the first and last names seems to have been mixed. It should be “Elgered, G; Johansson, J. M., Rönnäng, B. O.”. Check the other references for similar mistakes.
Response 14: We had checked all the literature author information according to your prompts and corrected the wrong author information.
